# SimCLIP: Refining Image-Text Alignment with Simple Prompts for Zero-/Few-shot Anomaly Detection

## ABSTRACT

Recently, large pre-trained vision-language models, such as CLIP, have demonstrated significant potential in zero-/few-shot anomaly detection tasks. However, existing methods not only rely on expert knowledge to manually craft extensive text prompts but also suffer from a misalignment of high-level language features with fine-level vision features in anomaly segmentation tasks. In this paper, we propose a method, named SimCLIP, which focuses on refining the aforementioned misalignment problem through bidirectional adaptation of both Multi-Hierarchy Vision Adapter (MHVA) and Implicit Prompt Tuning (IPT). In this way, our approach requires only a simple binary prompt to accomplish anomaly classification and segmentation tasks in zero-shot scenarios efficiently. Furthermore, we introduce its few-shot extension, SimCLIP+, integrating the relational information among vision embedding and skillfully merging the cross-modal synergy information between vision and language to address AD tasks. Extensive experiments on two challenging datasets prove the more remarkable generalization capacity of our method compared to the current state-of-the-art. Our code and pre-trained models are available at https://anonymous.4open.science/r/SimCLIP-CAEC.

## CCS CONCEPTS

• **Computing methodologies → Scene anomaly detection**.

## KEYWORDS

Vision-Language Model, Vision Adapter, Prompt Learning

## 1 INTRODUCTION

One significant disparity between artificial intelligence and humans lies in their abilities to generalize novel tasks with limited data. Addressing anomaly detection (AD) tasks with limited data is a highly challenging and non-trivial problem, primarily due to the diversity of anomaly types and the scarcity of abnormal samples. Recent advancements in Vision-Language (V-L) models (e.g., CLIP [30] ) have shown promising capabilities in zero-shot AD tasks by effectively aligning natural language with visual information during pretraining. However, the major gap between the initial CLIP task setting and the typical anomaly detection results in poor semantic alignment across vision and language modalities on the

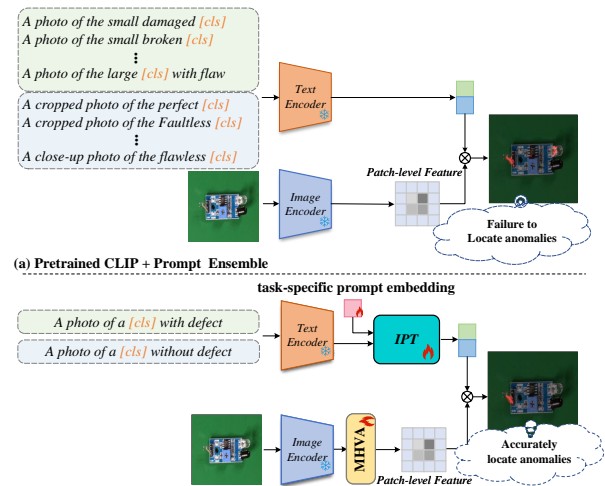

(a) Pretrained CLIP + Prompt Ensemble

(b) Pretrained CLIP + Implicit Prompt Tuning (IPT) +Multi-Hierarchy Vision Adapter (MHVA)

**Figure 1: (a) Prior works rely on the extensive hand-crafted text prompt and locate anomalies through the similarity between fine-level image patches and high-level prompt ensemble. (b) Our proposed SimCLIP enables realignment through bidirectional coordination between Implicit Prompt Tuning (IPT) and Multi-Hierarchy Vision Adapter (MHVA).**

AD task. Therefore, applying CLIP directly to zero-/few-shot AD tasks inevitably encounters two challenges.

On the one hand, the strong generalization of CLIP relies on text prompts aligned to images. In real-world scenarios, the diverse types of anomalies make it challenging for general text prompts to encompass them comprehensively. Existing approaches [6, 17] mitigate this problem by integrating expert knowledge and manually crafting extensive prompts (Figure 1(a)). However, its effectiveness is the necessity for verification with a certain degree of prior knowledge. Besides, creating substantial hand-crafted task-specific prompts for each scenario is time-consuming. Recent prompt learning works (e.g., CoOp [49]) can address this by directly learning prompts from training data of downstream tasks. Such methods can obtain better prompts in contrast to hand-crafted ones, but the learned prompts are bounded by the distribution associated with training data and have limited generalization [35]. In addition, the learned prompts through this approach lack interpretability and semantic coherence, which contradicts the original intention of CLIP aligning image-text pairs. This inspires us to explore an approach that can effectively guide the CLIP in accomplishing AD tasks using only simple prompts.

On the other hand, CLIP is an image-text aligning Pre-trained model well-suited for classification tasks. Nonetheless, AD encompasses the task of Anomaly Segmentation (AS), demanding pixel-level localization of anomalies. To accomplish AS task, current

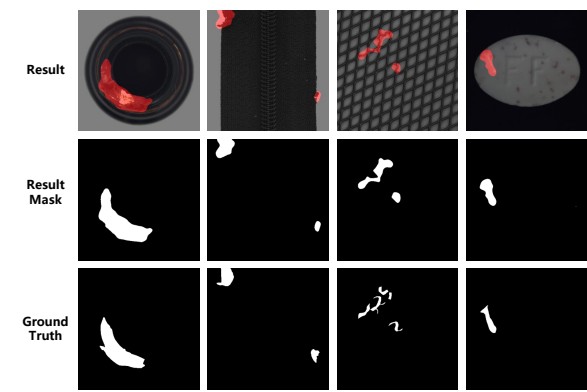

**Figure 2: Qualitative results of SimCLIP in zero-shot anomaly segmentation.**

methods[6] use linear mapping to transform patch-level features into a multi-modal feature space. Subsequently, AS is achieved by calculating the similarity of features between image patches and text prompts in this embedding space (Figure 1(a)). Due to the absence of changes in the prompt features, this process entails adapting fine-level patch features to high-level semantic prompt features. However, this unidirectional adaptation is insufficient to fully exploit the potential of CLIP in AS tasks. This insight motivates an exploration into the question: *How to effectively align patch features with text features in the multi-modal feature space?*

In this paper, we propose a novel approach, named SimCLIP, designed to bridge the gap between CLIP and downstream AD tasks in zero-/few-shot scenarios. As shown in Figure 1(b), SimCLIP accomplishes realignment of vision and language through bidirectional interaction adjustments of both two branches. For the vision branch, SimCLIP employs a multi-hierarchy vision adapter (MHVA) at different hierarchies to capture richer spatial information and enable effective cross-modal interactions with language. In the language branch, we leverage task-specific prompt embedding to learn relevant knowledge for AD tasks, alongside employing implicit prompt learning (IPT) to refine and optimize the original prompt. SimCLIP presents the following advantages: 1) Cross-modal realignment significantly enhances the performance of the pre-trained CLIP model in downstream AD tasks, demonstrating superior zero-shot generalization compared to the initial CLIP, as shown in Figure 2. 2) By employing prompt learning within a latent embedding space to refine initial prompts, we overcome the limitations imposed by text information capacity and broaden the solution space. 3) Implicit prompt learning significantly reduces the dependency on numerous hand-crafted prompts typically required for V-L models, enhancing the flexibility and efficiency of the model. 4) SimCLIP maintains the interpretability and semantic coherence of the text by fusion simple prompts with task-specific prompts in embedding space. Furthermore, we introduce SimCLIP+, an extended version of SimCLIP, integrating the relational information among vision features and merging the cross-modal synergy information between vision and language to address few-shot AD tasks.

In summary, we make the following main contributions:

- We propose a novel zero-shot AD approach named SimCLIP, which achieves realignment on pre-trained CLIP through bidirectional adjustment of both the multi-hierarchy vision adapter (MHVA) and implicit prompt learning (IPT). This method simplifies prompt design and seamlessly adapts to anomaly detection tasks involving unseen classes.
- To tackle AD tasks under limited samples, we introduce an extended version of SimCLIP, called SimCLIP+. This method not only utilizes intrinsic correlation information among vision embeddings but also integrates the cross-modal synergy information generated by SimCLIP. To further enhance anomaly detection efficiency, we also propose a prior-aware optimization algorithm designed to optimize the cross-modal synergy information mentioned above.
- We conduct extensive experiments on MVTec-AD [2] and VisA [52] benchmarks. SimCLIP/SimCLIP+ achieves superior performance for zero-/few-shot anomaly classification and segmentation compared to the state-of-the-art.

## 2 RELATED WORK

### 2.1 Vision-Language Models

The Vision-Language models have attracted significant interest in the field of computer vision through the integration of language supervision with images. Recent V-L models [18, 30] employ joint learning of two branches to connect these two modalities. They further align both the vision and language in the multi-modal feature space. Compared to models solely trained through vision supervision, these V-L models encode richer multi-modal representations. During the training process, these models can more effectively comprehend the natural world by leveraging information from both modalities, contributing to their outstanding performance across a wide range of tasks, including those involving zero-shot or few-shot visual recognition tasks. While these pre-trained V-L models learn generalized representations, effectively fine-tuning them to adapt to specific downstream tasks remains a challenging issue. Many studies have achieved state-of-the-art performance in downstream tasks, including image recognition[1, 26], object detection[12, 29, 36, 41], and segmentation[4, 9, 27, 32, 42, 48], by employing customized approaches to adapt V-L models for these specific tasks.

### 2.2 Prompt Learning

Prompt learning originated in the field of Natural Language Processing (NLP). Large Language Models (LLMs) [3, 7, 8, 22, 38, 46] acquire extensive knowledge during the pre-training phase by employing self-supervised learning methods on massive corpora. Prompt learning [10, 15, 21, 23, 34, 39] was developed to bridge the gap between Pre-trained Language Models (PLMs) and specified downstream tasks. For example, when recognizing the emotion of a social media post, "I got fired today," we may continue with a prompt "I felt so [blank]" and ask the PLMs to fill the blank with an emotion-bearing word. In this way, prompt learning may close the gap between PLMs and downstream tasks by converting downstream tasks into fill-in-blank tasks. Inspired by prompt learning in NLP, the paradigm of prompt learning has recently gradually expanded to encompass other domains, including V-L models [11, 19, 20, 45, 50]. In this paper, we explore a crucial issue: how to mitigate the gap between the CLIP [30] and AD tasks with prompt learning.

## 2.3 Zero-/Few-shot Anomaly Detection

Recent anomaly detection research has primarily concentrated on addressing anomaly classification and localization tasks with an extremely limited number of normal images. RegAD [16] employs image alignment tasks to train a category-agnostic anomaly detection model. WinCLIP [17] needs the hand-crafted design of textual prompts for both normal and anomalous conditions, and anomalies are identified by calculating the cosine similarity between textual prompts and images. While WinCLIP exhibits excellent performance, its effectiveness is heavily influenced by the carefully hand-crafted textual prompts. In a practical industrial setting, the meticulous design of textual prompts for each category is a tedious and time-consuming task. AnomalyGPT [13] takes image features as input for the LLM, leveraging the knowledge of the LLM to capture anomalous patterns within the images. Due to the core reliance on LLM, AnomalyGPT introduces a considerable computational overhead during the inference stage, which is unfavorable for real-time applications.

## 3 METHOD

In this section, we begin with a brief overview of CLIP [30], which serves as the foundational model employed in our method. Subsequently, we elaborate on SimCLIP, employing a multi-hierarchy vision adapter (MHVA) and implicit prompt tuning (IPT) to achieve realignment between vision and language. Finally, we propose an extended version of SImCLIP, named SimCLIP+, to address few-shot AD tasks by integrating a feature-driven method and a prior-aware optimization algorithm.

## 3.1 Preliminaries

CLIP consists of two branches: the vision branch $\mathcal{I}$ is dedicated to capturing the visual features of images, and the language branch $\mathcal{T}$ converts text prompts into semantic embeddings. During the pre-training phase, CLIP employs contrastive learning [5, 14] to maximize the cosine similarity between vision and language features that correspond to the same semantics within the multi-modal feature space. This is intended to promote the alignment between visual and textual features. Thus, CLIP can effectively leverage the distance between two modalities to accomplish zero-shot recognition tasks. Let $x$ and $\{P_j\}_{j=1}^K$ denote the inputs to the vision branch $\mathcal{I}$ and the language branch $\mathcal{T}$ respectively. Each prompt $P_j$ corresponds to a category, with $K$ being the total number of categories assumed. Specifically, each $P_j$ is obtained through a specific prompt template, such as "a photo of a {class}," where the "{class}" token is replaced with the name of the $j$-th class. The prediction probability is then calculated as:

$$p(y = j|x) = \frac{\exp(\text{sim}(\mathcal{I}(x), \mathcal{T}(P_j))/\tau)}{\sum_{i=1}^K \exp(\text{sim}(\mathcal{I}(x), \mathcal{T}(P_i))/\tau)}, \quad (1)$$

where $\text{sim}(\cdot, \cdot)$ denotes cosine similarity and $\tau$ is a temperature coefficient.

## 3.2 SimCLIP for zero-shot AD

In this section, we introduce SimCLIP, a method designed to efficiently drive CLIP to perform zero-shot AD tasks by leveraging a combination of the multi-hierarchy vision adapter and implicit prompt tuning. This approach aims to realign the vision and language modality within the multi-modal feature space.

CLIP focuses exclusively on achieving precise alignment between high-level global semantic information extracted from images and text during the pretraining phase. However, the extraction of high-level global semantic information often results in the loss of spatial details, thereby impeding the accurate localization of anomalies within images.

To overcome this limitation, as shown in Figure 3, we extract patch-level feature maps at various hierarchies. Formally, let $L$ represent the subset containing the indexes of the hierarchies to be utilized. The feature maps of the $l \in L$ are represented as $\mathcal{F}_{l,i} \sim \mathcal{I}_l(x_i) \in \mathbb{R}^{H_l \times W_l \times C_l}$, where $x_i \in \mathcal{D}_{train} \cup \mathcal{D}_{test}$ denotes the input image. Here $H_l$, $W_l$, and $C_l$ refer to the height, width, and depth of the feature map, respectively. We then select a feature slice at the spatial position $h \in \{1, 2, \ldots, H_l\}$ and $w \in \{1, 2, \ldots, W_l\}$ from the $l$-th layer's feature map, denoted by $\mathcal{F}_{l,i}(h, w) \sim \mathcal{I}_l(x_i, h, w) \in \mathbb{R}^{C_l}$, which is a $C_l$-dimensional real vector. To enable cross-modal interaction with text prompts and retain spatial information within the patch-level feature maps from different levels, we introduce a multi-hierarchy vision adapter (MHVA) $\mathcal{A}_{l,\theta}$. The adaptation process is represented as follows:

$$\mathcal{F}'_{l,i} = \mathcal{A}_{l,\theta}(\mathcal{F}_{l,i}(h, w)|\mathcal{F}_{l,i}(h, w) \in \mathcal{I}_l(x_i)), \text{where } l \in L. \quad (2)$$

We aim to maintain a design for the adapter that is both simple and efficient. Therefore, a linear probe is employed to construct the multi-hierarchy vision adapter. One choice for locating anomalies in an image is to calculate the feature similarity between the local regions of the input image (according to Eq.(2)) and text prompts. This is done in both [17] and [6] undoubtedly introduces the following two problems. Firstly, the abstract features extracted by the CLIP language branch are more tailored toward natural image classification tasks and hold limited relevance to segmentation tasks in industrial anomaly detection, with minimal overlap. Secondly, there is an inherent lack of alignment between the fine-level vision features and high-level language features. Directly utilizing these features would significantly degrade the model's generalization capability. In summary, this unidirectional vision adaptation method fails to exploit the potential of CLIP in AS tasks. Inspired by these insights, in addition to adapting the vision modality using the multi-hierarchy vision adapter, we also introduce implicit prompt tuning (IPT) to refine the language modality.

Firstly, we design simple binary prompts representing 'normal images' and 'anomalous images', respectively. In the remainder of this paper, unless explicitly indicated otherwise, the following binary prompts are the default:

$$P_0 = a\ photo\ of\ a\ [cls]\ without\ defect.$$
$$P_1 = a\ photo\ of\ a\ [cls]\ with\ defect. \quad (3)$$

Here, the symbol $[cls]$ denotes the category's name. Due to the limited capacity of text information, for instance, natural language is insufficient to encompass the entirety of anomaly types. Furthermore, to diminish reliance on hand-crafted prompts, we employ prompt learning to refine language features in latent embedding space. Let $T_{last,j} = \mathcal{T}(P_j) \in \mathbb{R}^{N \times C_{last}}$ denotes the output of the last transformer layer of the language branch, where $j \in \{0, 1\}$

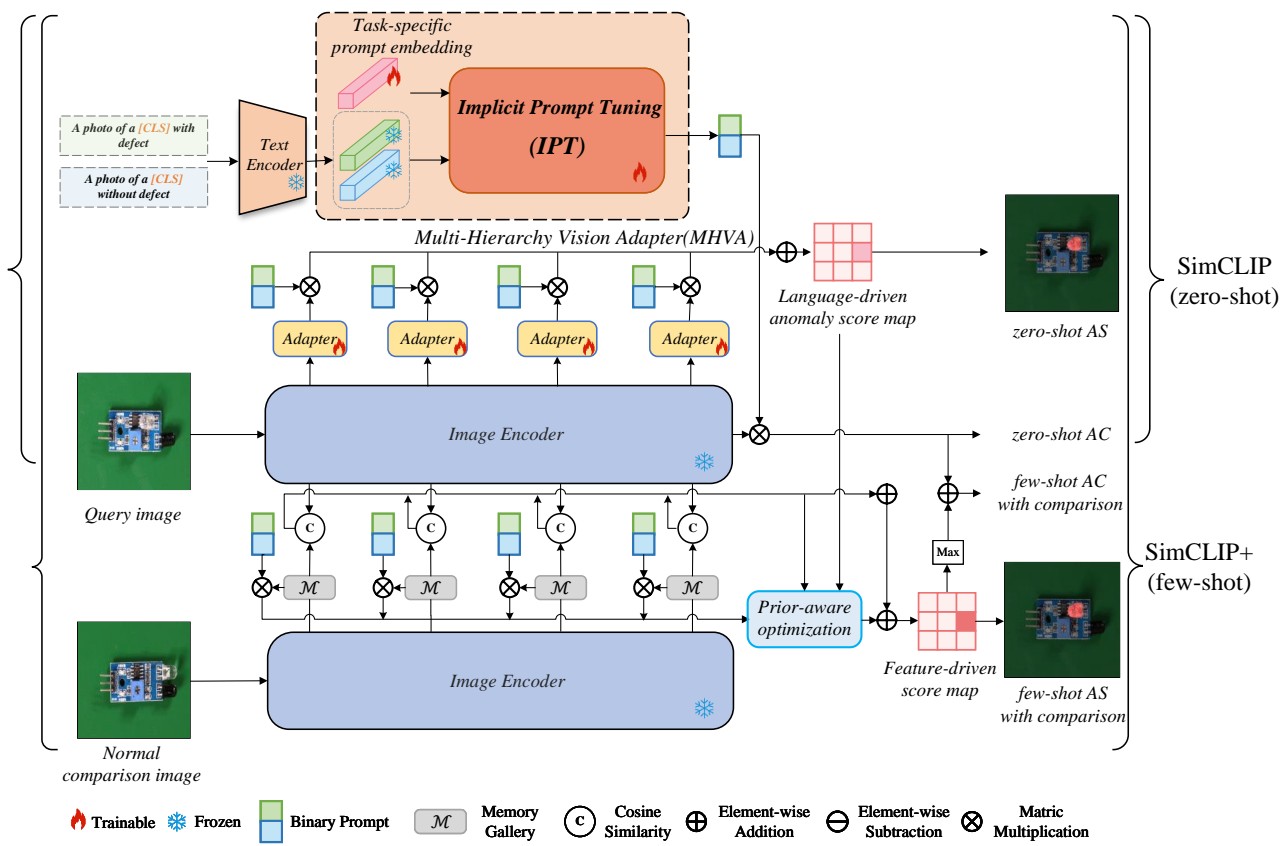

**Figure 3: The architecture of SimCLIP/SimCLIP+. For zero-shot AD, SimCLIP achieves realignment between vision and language through a cross-modal bidirectional adaptation approach. Otherwise, SimCLIP+ integrates a feature-driven method with prior-aware optimization algorithms to tackle few-shot AD tasks under limited normal samples.**

and $N$ denotes the number of tokens. As shown in Figure 3, we employ an implicit prompt tuning $\mathcal{L}_\theta$ integrated with learnable task-specific prompt embeddings $\mathcal{P}$ tailored for AD tasks to refine the original binary prompt. According to [24], it is evident that directly updating the $\mathcal{P}$ can result in unstable optimization processes and a slight performance decline. Therefore, we randomly initialize a relatively small matrix $\mathcal{P}_s$, then pass it through a single multi-layer perceptron (MLP) layer to address this limitation. The final task-specific prompt embedding can be obtained as follows:

$$\mathcal{P} = \text{MLP}(\mathcal{P}_s), \text{ where } \mathcal{P} \in \mathbb{R}^{C_{last}}. \quad (4)$$

As shown in Figure 3, the IPT $\mathcal{L}_\theta$ accepts prompt feature $T_{last,j}$ and task-specific prompt embedding $\mathcal{P}$ as inputs, consisting of two core components. Firstly, the weight transformation matrix $W = \{w^i\}_{i=1}^N$ accurately preserves key features from the original prompt while eliminating redundant information, where each weight $w^i \in [0, 1]$. Subsequently, the feedforward network (FFN) skillfully integrates the key features with task-specific prompt embeddings to attain refined prompts. The process of refining the text prompt through IPT is denoted as:

$$\mathcal{L}_\theta(T_{last,j}, \mathcal{P}) = \text{FFN}([T_{last,j}^\top W, \mathcal{P}]), \text{ where } j \in \{0, 1\}, \quad (5)$$

where $\mathcal{P}$ represents a new learnable task-specific prompt embedding in the latent feature space, and $[\cdot, \cdot]$ signifies the concatenation operation. Compared to other prompt learning methods[49, 50], IPT directly refines text prompts within the embedding space, expanding the solution space and markedly enhancing the flexibility of the learning process. Through the bidirectional adaptation facilitated by both MHVA and IPT for realigning both vision and language, the resulting anomaly segmentation outcome for language-driven at the $l$-th hierarchy can be generated as follows:

$$\mathbf{M}_{l,zero}(\mathcal{F}'_{l,i}) = \frac{\exp(\text{sim}(\mathcal{L}_\theta(T_{last,k=1}, \mathcal{P}), \mathcal{F}'_{l,i}))}{\sum_{j \in \{0,1\}} \exp(\text{sim}(\mathcal{L}_\theta(T_{last,j}, \mathcal{P}), \mathcal{F}'_{l,i}))}, \quad (6)$$

where $i$ denotes the $i$-th input image and $\mathcal{F}'_{l,i}$ represents the vision adaptation feature after the multi-hierarchy vision adapter. The final zero-shot AS result for SimCLIP is obtained by aggregating anomaly segmentation maps from different hierarchical levels, denoted as $\sum_{l \in L} \mathbf{M}_{l,zero}(\mathcal{F}'_{l,i})$. The zero-shot AC task can be accomplished by calculating the similarity between the vision features generated by the class token and the refined binary prompt.

**Notes.** In this paper, we finetune the pre-trained CLIP using the test set from a publicly available dataset (e.g., MVTec-AD) and then evaluate the performance using another dataset. All parameters of CLIP are frozen except for the projection head of both two branches. Additionally, parameter updates are performed on the newly introduced multi-hierarchy vision adapter, the task-specific prompt embedding, and the IPT module during the finetune process. We employ cross-entropy loss for AC, while focal loss [25] and dice loss [28] are utilized for AS tasks.

## 3.3 SimCLIP+ for few-shot AD

In this section, SimCLIP+ is proposed to tackle anomaly detection under limited samples, which leverages correlation information among vision embeddings and integrates the cross-modal synergy information generated by SimCLIP. Otherwise, SimCLIP+ also uses a prior-aware optimization algorithm to optimize the cross-modal synergy information mentioned above to further enhance the performance.

We address the challenge posed by limited normal samples $\mathcal{D}_N^+ = \{x_i\}_{i=1}^N$, where $N$ represents the number of normal images. Prior works [31, 37, 43, 44, 47] have extensively demonstrated the exceptional capability of CLIP in extracting features by vision branch. Motivated by these findings, we have decided to identify potential anomalies by measuring the distance at the embedding between the query and normal comparison images $x \in \mathcal{D}_N^+$. Building upon Section 3.2, where spatial information from various hierarchies is utilized, we introduce a *Memory Gallery* module $\mathcal{M}_l$ across different levels. The $\mathcal{M}_l$ aims to store patch-level features of all comparison images extracted from the $l$-th hierarchy on the vision branch, where $l \in L$ same to Section 3.2. As shown in Figure 3, leveraging the embedding distance between the query and comparison images at patch-level on various hierarchies, the anomaly segmentation result $\mathbf{M}_v$ generated using vision embedding information is represented as follows:

$$\mathbf{M}_v := \sum_{l \in L} \arg\min_{f \in \mathcal{M}_l} \frac{1}{2}(1 - \text{sim}(\mathcal{F}_{l,i}(h, w), f)). \qquad (7)$$

Here, $\mathcal{F}_{l,i}(h, w)$ denotes the feature slice of the $i$-th query image at $l$-th hierarchy. The feature-driven method leverages correlation information among vision embeddings and integrates the cross-modal synergy the information generated by SimCLIP. However, it overlooks the prior information between the refined text prompt and the normal comparison images. Inspired by these, we propose a prior-aware optimization algorithm, which is designed to further optimize the cross-modal synergy information mentioned above. Firstly, for each pixel-level feature $\mathcal{F}_{l,i}(h, w) \in I_l(x_i)$ retrieve the nearest neighbor from the $\mathcal{M}_l$, denoting as $f_{l,i}(h, w)$. The distance matrix $D_l$ is generated based on the similarity between $\mathcal{F}_{l,i}$ and $f_{l,i}$, where $D_l \in \mathbb{R}^{H_l \times W_l}$. Because of the substantial gap between anomaly features and their nearest neighbor, the value at this position in the distance matrix $D_l$ is smaller compared to other normal regions. Based on the analysis above, a pseudo-normal score map is generated by multiplying the language-guided normal score map $\mathbf{M}_{l,zero}(f_{l,i})$ with the distance matrix $D_l$. Finally, we utilize this

pseudo-normal score map $\mathbf{M}_{pseudo}$ to optimize cross-modal synergy information generated by SimCLIP, resulting in a new score map $\mathbf{M}_{zero}^{prior}$ that highlights anomalous regions more prominently.

The few-shot anomaly segmentation result is obtained by combining the $\mathbf{M}_v$ and $\mathbf{M}_{zero}^{prior}$. Lastly, we achieve few-shot anomaly classification by integrating the maximum value from the few-shot anomaly segmentation map with the language-guide zero-shot anomaly classification score.

---

**Algorithm 1** Prior-aware optimization algorithm

---

**Input:** $\forall x_i \in \mathcal{D}_{query}$, structure of $\mathcal{I}, \mathcal{T}, \mathcal{M}$
**for** $l \in L$ **do**
    **for** $\mathcal{F}_{l,i}(h, w) \in \mathcal{I}_l(x_i)$ **do**
        # find nearest neighbor feature
        $f_{l,i}(h, w) \leftarrow \text{NearesetNeighborAlgorithm}(\mathcal{F}_{l,i}(h, w), \mathcal{M}_l)$
    **end for**
    # calculate similarity to generate a distance matrix
    $\mathbf{D}_l \leftarrow \text{sim}(\mathcal{F}_{l,i}, f_{l,i})$
    $\mathbf{M}_{pseudo} \leftarrow \mathbf{D}_l \cdot \mathbf{M}_{l,zero}(f_{l,i})$
    # optimize cross-modal synergy information
    $\mathbf{M}_{l,zero}^{'}(\mathcal{F}_{l,i}) \leftarrow \mathbf{M}_{l,zero}(\mathcal{F}_{l,i}) - \mathbf{M}_{pseudo}$
**end for**
# aggregate anomaly score from various hierarchies
**Output:** $\mathbf{M}_{zero}^{prior} \leftarrow \sum_l^L \mathbf{M}_{l,zero}^{'}(\mathcal{F}_{l,i})$

---

## 4 EXPERIMENTS

We conducted a series of experiments to evaluate the effectiveness of SimCLIP/SimCLIP+ in industrial anomaly classification and anomaly segmentation tasks. In addition, a comprehensive ablation study is performed to verify the efficacy of each component proposed in our framework.

**Datasets.** Our experiments are conducted using the MVTec-AD [2] and VisA [52] datasets. MVTec-AD is composed of 15 subsets and encompasses a diverse range of defect types, such as scratches, dents, and contaminations, providing comprehensive coverage across various industrial sectors. VisA comprises 12 subsets, covering a range of structural anomalies such as misalignment or missing components, along with other defect types including cracks, corrosion, and more.

**Evaluation metrics.** For classification, we utilize the Area Under the Receiver Operating Characteristic curve (AUROC) and Average Precision (AP) as evaluation metrics. AUROC measures the trade-off between sensitivity and specificity, while AP considers the precision-recall trade-off. For segmentation, in addition to pixel-level AUROC, we also use Per-Region Overlap (PRO) as an essential metric in evaluating segmentation performance. In this paper, we use P-AUR and P-PRO to denote AUROC and PRO at the pixel level metric, respectively. Similarly, I-AUR and I-AP represent AUROC and PRO metrics at the image level.

**Implementation details.** We employ the pre-trained CLIP model developed by OpenAI, where the vision branch is based on the ViT (Vision Transformer) architecture. The utilized hierarchy $L = \{6, 12, 18, 24\}$. Training is conducted to one NVIDIA-3080Ti GPU over 5 epochs, using the Adam optimizer with a learning rate

**Table 1: Quantitative comparison of anomaly segmentation (AS) and anomaly classification (AC) performance on VisA and MVTec-AD datasets. We report the mean and standard deviation over 5 random seeds for each measurement. Bold indicates the best performance, while underline denotes the second-best result**

| Setup | Method | VisA | | | | MVTec-AD | | | |
|---|---|---|---|---|---|---|---|---|---|
| | | P-AUR | P-PRO | I-AUR | I-AP | P-AUR | P-PRO | I-AUR | I-AP |
| **0-shot** | CLIP-AC | 47.8±0.0 | 17.3±0.0 | 65.0±0.0 | 70.1±0.0 | 38.2±0.0 | 11.6±0.0 | 71.5±0.0 | 86.4±0.0 |
| | APRIL-GAN | 94.2±0.0 | 86.8±0.0 | 78.0±0.0 | 81.4±0.0 | 87.6±0.0 | 44.0±0.0 | 86.1±0.0 | 93.5±0.0 |
| | AnomalyCLIP | 95.5±0.0 | 87.0±0.0 | 82.1±0.0 | 85.4±0.0 | 91.1±0.0 | 81.4±0.0 | **91.5±0.0** | **96.2±0.0** |
| | SimCLIP(ours) | **95.6±0.0** | **89.7±0.0** | **83.1±0.0** | **86.0±0.0** | **91.8±0.0** | **86.8±0.0** | 90.0±0.0 | 95.3±0.0 |
| **1-shot** | PatchCore | 95.4±0.6 | 64.3±2.4 | 79.9±2.9 | 82.8±2.3 | 93.3±0.6 | 82.3±1.3 | 86.3±3.3 | 92.2±1.5 |
| | RegAD | 93.6±0.2 | 72.0±0.5 | 68.4±1.0 | 71.2±0.5 | 92.8±0.5 | 77.9±1.2 | 76.5±2.0 | 88.2±0.9 |
| | APRIL-GAN | 96.0±0.0 | 90.0±0.1 | 91.2±0.8 | 93.3±0.8 | 95.1±0.1 | 90.6±0.2 | 92.0±0.3 | 95.8±0.2 |
| | AnomalyGPT | 96.2±0.1 | - | 87.4±0.8 | - | 95.3±0.1 | - | 94.1±1.1 | - |
| | SimCLIP+(ours) | **97.4±0.1** | **92.7±0.2** | **93.0±1.1** | **94.5±0.9** | **95.6±0.2** | **92.4±0.2** | **95.3±0.3** | **97.7±0.3** |
| **2-shot** | PatchCore | 96.1±0.5 | 82.6±2.3 | 81.6±4.0 | 84.8±3.2 | 92.0±1.0 | 79.7±2.0 | 83.4±3.0 | 93.8±1.7 |
| | RegAD | 94.4±0.3 | 73.4±0.8 | 73.3±1.4 | 75.0±0.7 | 94.6±0.3 | 86.3±0.9 | 85.7±1.3 | 92.7±0.7 |
| | APRIL-GAN | 96.2±0.0 | 90.1±0.1 | 92.2±0.3 | 94.2±0.3 | 95.5±0.0 | 91.3±0.1 | 92.4±0.3 | 96.0±0.2 |
| | AnomalyGPT | 96.4±0.1 | - | 88.6±0.7 | - | 95.6±0.2 | - | 95.5±0.8 | - |
| | SimCLIP+(ours) | **97.7±0.1** | **93.4±0.0** | **93.7±0.2** | **94.9±0.2** | **96.0±0.2** | **92.9±0.1** | **96.0±0.2** | **98.1±0.1** |
| **4-shot** | PatchCore | 96.8±0.3 | 84.9±1.4 | 85.3±2.1 | 87.5±2.1 | 94.3±0.5 | 84.3±1.6 | 88.8±2.6 | 94.5±1.5 |
| | RegAD | 95.9±0.2 | 76.5±0.9 | 73.8±0.8 | 75.8±1.8 | 95.8±0.3 | 88.1±0.8 | 88.2±1.3 | 94.8±0.6 |
| | APRIL-GAN | 96.2±0.0 | 90.2±0.1 | 92.6±0.4 | 94.5±0.3 | 95.9±0.0 | 91.8±0.1 | 92.8±0.2 | 96.3±0.1 |
| | AnomalyGPT | 97.2±0.2 | - | 90.6±0.7 | - | **96.2±0.1** | - | 96.3±0.3 | - |
| | SimCLIP+(ours) | **98.0±0.2** | **94.1±0.1** | **94.4±0.1** | **95.6±0.1** | **96.2±0.1** | **93.1±0.1** | **96.4±0.2** | **98.0±0.2** |

of $1e-3$ to update model parameters. We finetune SimCLIP on the MVTec-AD dataset and evaluate its generalization performance on Visa. Similarly, we finetune SimCLIP on the VisA dataset and evaluate its generalization performance on MVTec-AD.

## 4.1 Zero-shot anomaly detection

We assess the performance of SimCLIP for zero-shot anomaly detection tasks using two benchmark datasets and conduct a comparative analysis with CLIP-AC[30], APRIL-GAN[6] and AnomalyCLIP[51]. The result of CLIP-AC in anomaly segmentation is poor because the original CLIP only focuses on high-level global semantic information extracted from images and text during the pretraining phase. APRIL-GAN has achieved promising performance by integrating prompt ensemble and features in its approach. AnomalyCLIP accomplishes better performance by learning object-agnostic prompts. On the VisA, SimCLIP demonstrates superior efficacy, surpassing the second-ranked AnomalyCLIP by 1%/0.6% in AUROC/AP for anomaly classification. On the MVTec-AD, while SimCLIP exhibits inferior anomaly classification compared to AnomalyCLIP, there is an improvement of 0.7% in AUROC and 5.4% in PRO for anomaly segmentation. SimCLIP focuses on realignment between vision and language, enabling the capture of subtle differences in images, and contributing to its better performance in anomaly segmentation tasks.

## 4.2 Few-shot anomaly detection

We conducted comprehensive comparisons and analyses with methods including PatchCore[33], AnomalyGPT[13], RegAD[16], APRIL-GAN on two benchmark datasets for both few-shot anomaly segmentation and anomaly classification. PatchCore and RegAD, which concentrate exclusively on vision detection without incorporating multi-modal information, demonstrate performance constraints that impact their competitiveness in few-shot anomaly detection. APRIL-GAN achieves better performance by manually crafting a large set of text prompts and ensemble them to guide V-L models. AnomalyGPT achieves solid outcomes in anomaly segmentation by effectively leveraging a prompt learner to fine-tune large vision-language models. SimCLIP+ integrates the relational information among vision features and merges the cross-modal synergy information between vision and language, achieving the best result. In 1-/2-/4-shot anomaly classification, SimCLIP+ exhibits stronger performance when compared to AnomalyGPT, showing enhancements of 5.6%/5.1%/3.8% in AUROC respectively on the Visa benchmark.

## 4.3 IPT vs. SOTA prompt learning methods

Table 4 reports a comparison of the implicit prompt tuning in Sim-CLIP with current state-of-the-art prompt learning methods in zero-shot anomaly segmentation and anomaly classification. We

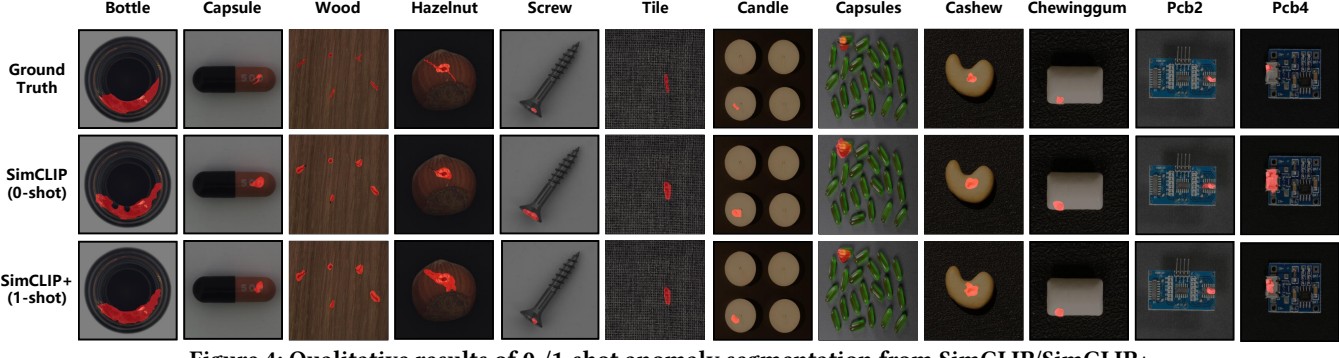

**Figure 4: Qualitative results of 0-/1-shot anomaly segmentation from SimCLIP/SimCLIP+.**

**Table 2: Comparison with existing prompt learning methods on both computation and memory overhead.**

| Method | FLOPs(G) | Params(M) |
|---|---|---|
| Coop | 520.46 | 428.79 |
| Co-CoOp | 520.46 | 428.92 |
| IPT(ours) | 513.75 | 428.77 |

**Table 3: Ablation on two key modules of SimCLIP on the VisA dataset. Bold indicates the best performance.**

| MHVA | IPT | (P-AUR, P-PRO) | (I-AUR, I-AP) |
|---|---|---|---|
| | | (22.9, 11.7) | (67.4,72.2) |
| ✓ | | (94.3,84.0) | (67.4,72.2) |
| | ✓ | (22.3,11.3) | (83.0,85.7) |
| ✓ | ✓ | **(95.6,89.7)** | **(83.1,86.0)** |

**Table 4: Generalization comparison of IPT with existing prompt learning methods on both anomaly segmentation and anomaly classification tasks. Bold indicates the best performance.**

| Dataset | Method | (P-AUR, P-PRO) | (I-AUR, I-AP) |
|---|---|---|---|
| **Visa** | Coop | (93.9,75.1) | (74.7,79.0) |
| | Co-CoOp | (93.8,73.6) | (79.5,82.5) |
| | IPT(ours) | **(95.6,89.7)** | **(83.1,86.0)** |
| **MVTec-AD** | Coop | (79.2,26.1) | (87.4,94.4) |
| | Co-CoOp | (81.1,40.9) | (84.4,91.7) |
| | IPT(ours) | **(91.8,86.8)** | **(90.0,95.3)** |

conduct experiments within the SimCLIP framework, replacing the implicit prompt tuning with other SOTA prompt learning methods to ensure a fair comparison. The implicit prompt tuning achieves the best performance in both anomaly segmentation and anomaly classification on MVTec-AD, with improvements exceeding 10.7% in pixel-level AUROC and 5.6% in image-level AUROC compared to Co-CoOp. Similarly, this trend extends to the VisA. In comparison with CoOp and Co-CoOp, which obtain text prompts through learnable context optimization and conditional context optimization based on an image, implicit prompt tuning offers the following two advantages. Firstly, unlike methods that optimize at the input side of the language branch, implicit prompt tuning optimizes the text prompt at the output side. Not only does IPT avoid the biases introduced by the language branch, which might overly focus on foreground objects in the image rather than anomalies, but it also improves optimization speed by not requiring the gradient to backpropagate through the entire language branch. Secondly, implicit prompt tuning effectively complements domain knowledge by introducing task-specific prompt embeddings for anomaly detection tasks, enhancing the model's context understanding and discriminative capability.

In addition to comparing the performance between IPT and SOTA prompt learning methods, we also assessed their differences in computational and memory overhead. As shown in Table 2, With a similar number of parameters to other methods, IPT requires fewer Flops, resulting in higher computational efficiency.

## 4.4 Ablation study

**Multi-Hierarchy Vision Adapter and Implicit Prompt Tuning.** We focus on investigating the effectiveness of two critical modules on the generalization performance of pre-trained CLIP. From this perspective, we start by removing the MHVA and IPT modules, optimizing only the projection of the vision and language branches during fine-tuning, which serves as our baseline. Subsequently, the MHVA and IPT modules are integrated by us separately into the baseline to compare unidirectional with bidirectional adaptation.

Based on the baseline results as shown in Table 3, it is evident that the original CLIP is more adept at handling AC tasks. However, the performance in downstream AS tasks is somewhat underwhelming, further highlighting the gap between CLIP and downstream AD tasks. Although unimodal adaptation of either MHVA or IPT to some extent alleviates this issue, the potential of CLIP has not been fully realized. By employing cross-modal realignment through bidirectional adjustment of both the MHVA and IPT modules, there was an improvement of 72.7% and 15.7% in AUROC for AS and AC tasks compared to the baseline, respectively.

**Table 5: Ablation on multi-hierarchies vision adapter on the VisA dataset. Bold indicates the best performance.**

| Hierarchy | Hierarchy1 | Hierarchy2 | Hierarchy3 | Hierarchy4 | All |
|---|---|---|---|---|---|
| (P-AUR, P-PRO) | (89.7, 74.1) | (93.8, 83.5) | (94.7, 85.8) | (94.6, 85.8) | (**95.6,89.7**) |

**Table 6: Ablation on two key modules of SimCLIP+ on the VisA dataset and (P-AUC, I-AUC) is used as evaluation metrics. FD and PAO denote the feature-driven method and prior-aware optimization algorithm, respectively. Bold indicates the best performance.**

| FD | PAO | 1-shot | 2-shot | 4-shot |
|---|---|---|---|---|
| ✓ |  | (97.2,83.6) | (97.5,87.8) | (97.6, 92.1) |
|  | ✓ | (96.5, 84.0) | (96.6,84.0) | (96.6, 84.2) |
| ✓ | ✓ | (**97.4,93.0**) | (**97.7,93.7**) | (**98.0,94.4**) |

**Adapter Ensemble for Multi-Hierarchy.**In this study, we investigate the effectiveness of ensemble visual adapters across various hierarchies. We conduct a set of control experiments comprising two different conditions: 1)Using a multi-hierarchy vision adapter same as SimCLIP. 2)Using only a single-hierarchy vision adapter where the hierarchy is a subset of the multi-hierarchy level. Additionally, pixel-level AUROC and PRO are used as performance evaluation metrics in our study. As shown in Table 5, the multi-hierarchy vision adapter outperforms the other four single-hierarchy vision adapters, demonstrating its effectiveness. Compared to the second-best performing single-hierarchy vision adapter, the multi-hierarchy vision adapter shows improvements of 0.9% and 3.9% in pixel-level AUROC/PRO, respectively.

**Feature-driven method and prior-aware optimization algorithm.**Furthermore, we explore the effectiveness of the feature-driven method and the prior-aware optimization algorithms within SimCLIP+ in the few-shot settings. In this study, We consider three scenarios: 1)Using single feature-driven methods. 2)Using single prior-aware optimization algorithms. 3)Integrating feature-driven methods with prior-aware optimization algorithms at the same time. Table 6 reports the ablation result, using either a feature-driven method or a prior-aware optimization algorithm alone showing promising results in anomaly segmentation tasks. However, for anomaly classification tasks, these single methods often struggle to achieve accurate classifications. Combining feature-driven methods with prior-aware optimization algorithms leverages both vision embedding and cross-modal information, significantly enhancing anomaly segmentation and anomaly classification performance compared to using either method alone in the 1-/2-/4-shot settings.

### 4.5 Visualization Analysis

Figure 4 reports the visualization results of SimCLIP/SimCLIP+ on two benchmark datasets. Through realignment of vision and language, SimCLIP can accurately locate anomalies in images, including the ability to detect multiple anomalies within a single

image(e.g., the 'Wood' in Figure 4). The qualitative analysis demonstrates that SimCLIP+ outperforms SimCLIP in anomaly segmentation. This indirectly highlights the effectiveness of SimCLIP+ in integrating both vision embedding information and cross-modal synergy information concurrently.

As shown in Figure 5, We employ t-SNE [40] to visualize the distances between normal prompt and anomalous prompt embedding in the feature space. Figure 5(a) shows that there are no distinct boundaries separating the different types of text prompts in the feature space. This severely hinders CLIP's application in downstream anomaly detection tasks. The effectiveness of IPT is visually demonstrated by the clear boundaries observed between different types of text prompts in the feature space after refining, as illustrated in Figure 5(b).

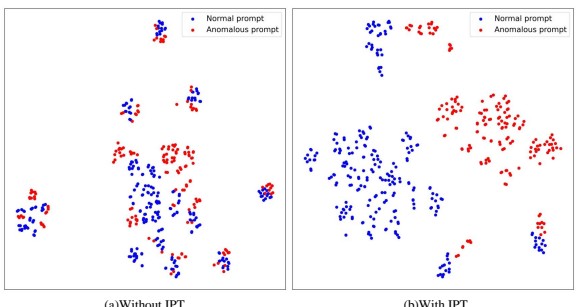

(a)Without IPT  (b)With IPT

**Figure 5: Visualization of normal/anomalous prompt features using t-SNE. (a) Original text prompt in feature space. (b) Refined text prompt (using IPT) in feature space.**

## 5 CONCLUSION

In this paper, we propose a novel method, named SimCLIP that employs bidirectional adaptation to accomplish realignment of vision and language, enhancing CLIP's zero-shot generalization performance on downstream AD tasks. In the vision branch, we incorporate a multi-hierarchy vision adapter situated at various levels to capture more intricate spatial details and facilitate efficient cross-modal interactions with language. In the language branch, we employ a learnable task-specific prompt embedding and the implicit prompt tuning module to refine the original prompts. SimCLIP bridges the gap between CLIP and downstream zero-shot AD tasks by bidirectionally adjusting both two branches. Additionally, we further propose SimCLIP+, which integrates correlation information among vision embeddings with cross-modal synergy information, coupled with a prior-aware optimization algorithm to address AD tasks under limited normal samples. Our proposed method provides a new perspective on bridging the gap between pre-trained vision-language models and downstream zero/few-shot AD tasks.

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
