# OpenReview forum: "SimCLIP: Refining Image-Text Alignment with Simple Prompts for Zero-/Few-shot Anomaly Detection"
_acmmm.org/ACMMM/2024/Conference — MM2024 Poster_

### Official Review · Reviewer_CdXm · 2024-05-18

**Rating:** 4
**Confidence:** 2

**Summary:**

The paper proposed a SimCLIP that employs bidirectional adaptation to accomplish realignment of vision and language, enhancing CLIP’s zero-shot generalization performance on downstream AD tasks. The experiments on two challenging datasets verified the effectiveness of the proposed method.

**Strengths:**

(1) This paper explores the zero-shot generalization performance of enhanced CLIP on downstream AD tasks. The proposed method is a promising approach.
(2)The presented method is technically sound. Its key components are clearly explained with clear formulations.
(3) The experiments conducted on the MVTec-AD and VisA datasets demonstrate the effectiveness of the proposed method.

**Limitations:**

（1）The author should carefully proofread the paper as there are several issues with details. 1) There are inconsistencies in descriptions, such as 'Datasets.Our' at line 557; 2) Table 1 is not cited in the text; 3) Some paragraphs are too long, so it is suggested to divide them without changing the content.
（2）The lack of introduction and analysis on recent relevant work [1-2] is noticeable.
[1] Li Y, Goodge A, Liu F, et al. PromptAD: Zero-Shot Anomaly Detection Using Text Prompts[C]//Proceedings of the IEEE/CVF Winter Conference on Applications of Computer Vision. 2024: 1093-1102.
[2] Gu Z, Zhu B, Zhu G, et al. Anomalygpt: Detecting industrial anomalies using large vision-language models[C]//Proceedings of the AAAI Conference on Artificial Intelligence. 2024, 38(3): 1932-1940.
（3）Sections 4.1 and 4.2 describe some results without explicitly indicating which corresponding table of experimental data supports these outcomes.
（4）Evaluation indicator for the experiment is insufficient.
（5）The authors do not clearly articulate the advantages of the proposed method compared to existing methods.
（6）Experiment lacks comparison of computational complexity and inference time.

**Suitability:**

2

---

### Official Review · Reviewer_kV7g · 2024-05-24

**Rating:** 4
**Confidence:** 3

**Summary:**

To deal with the misalignment of high-level language features with fine-level vision features in anomaly segmentation tasks for large pre-trained vision-language models, this paper proposes the SimCLIP to refine the misalignment. Specifically, SimCLIP employs a multi-hierarchy vision adapter at different hierarchies for the vision branch. For the language branch, SimCLIP employs implicit prompt learning to refine and optimize the original prompt. Besides, this paper introduces an extended version of SimCLIP, named SimCLIP+, to utilize intrinsic correlation information among vision embeddings and integrate the cross-modal synergy information generated by SimCLIP. Finally, extensive experiment results on MVTec-AD and VisA benchmarks demonstrate the robustness of the proposed SimCLIP and SimCLIP+.

**Strengths:**

1.	This paper is well-composed with a clear organizational structure. Figures 1, 2, and 3 effectively illustrate the motivation and detailed content of the paper, aiding in the comprehension of the author’s concepts and improving the overall readability for the audience.
2.	The paper provides the detailed algorithm pseudocode (i.e., as shown in Algorithm 1) and detailed code in the github link, demonstrating the reliability and reproducibility of the results.
3.	The experimental section provides qualitative and quantitative analyses, which intuitively and effectively demonstrate the performance of MHVA and IPT proposed for images and text branches, respectively, and validate the robustness of SimCLIP+.

**Limitations:**

1.	The authors mention in the paper that they design the simple binary prompts to refine the language modality. However, the paper lacks a deeper theoretical analysis and proof of this approach. Providing such analysis would enhance the credibility of the paper.
2.	The paper lacks ablation analysis and experimental results for the hyperparameters (e.g., \tau), and specific values are not provided in the experimental details section. The authors need to further supplement the corresponding experimental results.
3.	The experimental comparisons in the paper are insufficient. The authors only compare a few SOTA methods in Tables 1, 2, and 4, which is not enough to prove the superiority of the proposed method. The authors should provide more experimental comparison results with SOTA methods published in the last two years.
4.	The authors should provide specific indices corresponding to the specific parts of the figure in the caption of Figure 3 to facilitate readers' intuitive understanding of the proposed SimCLIP.

**Suitability:**

3

---

### Official Review · Reviewer_xcNW · 2024-05-24

**Rating:** 4
**Confidence:** 2

**Summary:**

The author proposes a method called SimCLIP, which improves the inconsistency between high-level language features and fine visual features through bidirectional adaptation of multi-level visual adapters and implicitly prompted tuning.

**Strengths:**

1. The paper has a clear motivation, the method is consistent with the motivation, and the paper is presented clearly.
2. The proposed method has achieved performance improvement on multiple datasets.

**Limitations:**

1. The model is trained using both the training set and another dataset. Will it cause leakage of new classes? I believe that ablation experiments should be added to obtain the performance results of models that do not use data from the other dataset.
2. Whether the methods compared in the paper also used additional datasets, and whether the backbone is consistent with this paper, I believe that these two points should be taken into consideration to ensure fairness. The authors can also try the results of other backbones.

**Suitability:**

3

---

### Official Review · Reviewer_SBQE · 2024-05-28

**Rating:** 3
**Confidence:** 3

**Summary:**

The manuscript presents a novel approach named SimCLIP, aimed at refining the misalignment between high-level language features and fine-level vision features in anomaly detection (AD) tasks. This is achieved through the bidirectional adjustment of both a Multi-Hierarchy Vision Adapter (MHVA) and Implicit Prompt Tuning (IPT). Additionally, an extended version, SimCLIP+, is introduced for few-shot scenarios, incorporating relational information among vision embeddings and cross-modal synergy. The results from extensive experiments on two challenging datasets, MVTec-AD and VisA, indicate superior performance in zero-/few-shot anomaly classification and segmentation compared to current methods.

**Strengths:**

1. The paper successfully integrates multiple advanced techniques, including MHVA and IPT, to address the alignment issues between vision and language modalities, which is a significant contribution to the field of anomaly detection.
2. The method simplifies the process of designing text prompts for AD tasks, potentially reducing the dependence on expert knowledge and making the approach more accessible.
3. The experiments results support for the effectiveness of the proposed approaches, SimCLIP and SimCLIP+.

**Limitations:**

1. This paper does not clarify whether there is a comparison of training costs and efficiency between SimCLIP/SimCLIP+ and other models, which is essential for evaluating practical deployment feasibility.

2. Parameter control is not specified if the comparisons of individual components in Table 3 control for total parameter count or computational cost, potentially affecting the accuracy of the comparative analysis.

3. There is a lack of performance data on more challenging datasets, which is necessary to fully assess the robustness and generalization capability of the proposed methods.

4.  The manuscript could be strengthened by more clearly highlighting distinctive innovations or unique aspects of SimCLIP and SimCLIP+ that distinguish them from existing methodologies.

**Suitability:**

3

---

### Meta-Review · Area_Chair_Gh4r · 2024-07-12

**Recommendation:** Accept (Poster)
**Confidence:** 5

**Metareview:**

This paper has been evaluated by 4 reviewers who all converge to the Borderline accept score. This score results from the merits of the paper and from the high quality elements found in the rebuttal, which helped reviewers to get a better understanding of some of the elements highlighted in the weaknesses sections of their evaluations. Authors make promises in the rebuttal -- that has to be implemented in the final version of the manuscript in order to improve the overall quality of the article, thanks to the issues highlighted by the reviewers.

All these reasons push the recommendation to be ACCEPT, poster.